# A succession of two viral lattices drives vaccinia virus assembly

**Miguel Hernandez-Gonzalez[1], Thomas Calcraft[2], Andrea Nans[3], Peter B Rosenthal[2], Michael Way**[1,4] *

**1** Cellular signalling and cytoskeletal function laboratory, The Francis Crick Institute, London, United Kingdom, **2** Structural Biology of Cells and Viruses Laboratory, The Francis Crick Institute, London, United Kingdom, **3** Structural Biology Science Technology Platform, The Francis Crick Institute, London, United Kingdom, **4** Department of Infectious Disease, Imperial College, London, United Kingdom

* michael.way@crick.ac.uk

## Abstract

During its cytoplasmic replication, vaccinia virus assembles non-infectious spherical immature virions (IV) coated by a viral D13 lattice. Subsequently, IV mature into infectious brick-shaped intracellular mature virions (IMV) that lack D13. Here, we performed cryo-electron tomography (cryo-ET) of frozen-hydrated vaccinia-infected cells to structurally characterise the maturation process in situ. During IMV formation, a new viral core forms inside IV with a wall consisting of trimeric pillars arranged in a new pseudohexagonal lattice. This lattice appears as a palisade in cross-section. As maturation occurs, which involves a 50% reduction in particle volume, the viral membrane becomes corrugated as it adapts to the newly formed viral core in a process that does not appear to require membrane removal. Our study suggests that the length of this core is determined by the D13 lattice and that the consecutive D13 and palisade lattices control virion shape and dimensions during vaccinia assembly and maturation.

## Introduction

Poxviruses are large double-stranded DNA viruses that replicate and assemble their virions in cytoplasmic perinuclear viral factories [1]. The family includes variola virus, the causative agent of smallpox, and monkeypox as well as vaccinia, the most studied family member that was used as the vaccine to eradicate smallpox [2,3]. During its replication cycle, vaccinia initially assembles into infectious intracellular mature virions (IMV), which are released when infected cells lyse [4]. Prior to cell lysis, however, some IMV become enveloped by a Golgi cisterna or endosomal compartment to form intracellular enveloped virions (IEV) [5–7] (Fig 1A). IEV undergo kinesin-1–mediated microtubule transport to the plasma membrane, where they fuse to release extracellular enveloped virions (EEV), which promotes the long-range spread of infection [8,9]. A proportion of these virions, named cell-associated enveloped virus (CEV), induce the formation of an actin tail beneath the virion, which increases the cell-to-cell spread of the virus [10–13].

The first step in the spread of vaccinia infection is the assembly of IMV, which are the precusor to all subsequent forms (IEV, EEV, and CEV) of the virus. IMV formation is initiated

**Data Availability Statement:** The maps from tomography (EMD-15600 showing immature virions (IV) and EMD-15601 showing intracellular mature virions (IMV)) and subtomogram averaging

(EMD-15596, EMD-15597, EMD-15598, EMD-15599, EMD-15602 for combined palisade, CEV/EEV palisade, IMV palisade, IEV palisade and D13 lattice respectively) have been deposited in the Electron Microscopy Data Bank. The docked atomic model of D13 has been deposited in the Protein Data Bank with the ID 8ARH. All other relevant data are within the paper and its Supporting information files.

**Funding:** This work was supported by the Francis Crick Institute, which receives its core funding from Cancer Research UK (CC2096 to MW; CC2106 to PBR), the UK Medical Research Council (CC2096 to MW; CC2106 to PBR), and the Wellcome Trust (CC2096 to MW; CC2106 to PBR). The funders had no role in study design, data collection and analysis, decision to publish, or preparation of the manuscript.

**Competing interests:** The authors have declared that no competing interests exist.

**Abbreviations:** ASFV, African swine fever virus; CEV, cell-associated enveloped virus; cryo-ET, cryo-electron tomography; CTF, contrast transfer function; EEV, extracellular enveloped virion; EFC, entry fusion complex; ER, endoplasmic reticulum; IEV, intracellular enveloped virion; IMV, intracellular mature virion; IV, immature virion; STA, subtomogram averaging; VMAP, viral membrane assembly protein.

when endoplasmic reticulum (ER)-associated viral membrane assembly proteins (VMAPs) promote the formation of isolated membrane crescents in the cytoplasm of infected cells [14–16]. The characteristic dimensions and shape of the crescents are determined and maintained by the D13 scaffolding viral protein, which forms trimers that assemble into a hexameric lattice coating the outer surface of these membranes [14,15,17,18]. In vitro structural analysis of recombinant D13 reveals it has a double-jelly-roll structure composed of 8 antiparallel β-strands that self assembles into a honeycomb lattice of pseudohexagonal trimers [19–22]. Ultimately, membrane crescents develop into spherical immature virions (IV) that encapsidate the viral genome and proteins required to produce an IMV [15,17,18,23]. In a process that is still not understood but involves the proteolytic cleavage of viral proteins and the formation of disulphide bonds, spherical IV undergo a dramatic reorganization into brick-shaped IMV [24,25]. During this maturation process, the D13 honeycomb lattice is lost, and a biconcave viral core, containing the viral genome, assembles inside the IV [23]. The outer layer of the viral core forms a paracrystalline palisade-like structure that has been reported to be discontinuous [23,26,27]. In addition, 2 dense proteinaceous aggregates, known as the lateral bodies, form in the cavity between the outer viral membrane and the biconcave viral core [23,26,28].

Vaccinia virion assembly has been extensively studied. Nevertheless, the key structural determinants that govern the changes occurring during the IV to IMV transistion, as well as those that define the IMV, remain unknown. To obtain detailed ultra-structural insights into this maturation process, we have now performed cryo-electron tomography (cryo-ET) of vaccinia-infected cells, imaging IV as well as infectious IMV in the cytoplasm. Using subtomogram averaging (STA), we determine the structure of the D13 lattice on the IV membrane. Moreover, STA of the IMV core reveals a continuous pseudohexagonal palisade lattice, which is reminiscent of a viral capsid. Our analysis also identifies novel features of the surrounding IMV membrane. We discuss a model for maturation from IV to IMV that is organised by the succession of the distinct D13 and palisade lattices.

## Results

### Electron cryo-tomography of vaccinia virus in infected cells

Cryo-tomograms of thin cellular regions containing IV and IMV were recorded from the periphery of HeLa cells infected with vaccinia virus for 8 h (Fig 1B and 1C, and S1 Fig and S1 and S2 Movies). Sample thickness varied depending on the region of the cell and the presence of a viral particle (Fig 1B and 1C and S1 Table). IV, fully coated by the viral D13 lattice, are largely spherical and have a diameter of 351.89 ± 2.88 nm (S2 Table). Analysis of tomograms in 3D, however, reveals that a proportion of IV despite being closed and fully coated by D13 have a single invagination of the viral membrane (Fig 1B). This suggests that the D13 lattice is flexible and can accommodate both positive and negative curvatures present on virions with invaginations. This D13 lattice flexibility is also evident from its ability to coat the exterior of "open spheres" with different membrane curvature (also known as crescents due to their shape when seen in a middle view) during IV formation (Fig 2A and 2B and S3 Movie).

The interior of the majority of IV, which contain viral DNA and proteins required for virion assembly, are homogenous and lack any of the distinctive regions or layers seen in IMV (Fig 1B). In a few cases, however, we could observe a single dense striated structure inside IV corresponding to the nucleoid (viral DNA genome and associated proteins) (Fig 2C and S2 Fig), as seen in previous ultra-structural analysis of thin EM sections of vaccinia-infected cells [28]. We also found examples of related striated structures in the cytoplasm adjacent to, or in direct association with, assembling IV (Fig 2D and 2E). Consistent with the interpretation that the example shown in Fig 2E represents the viral genome being inserted into assembling IV,

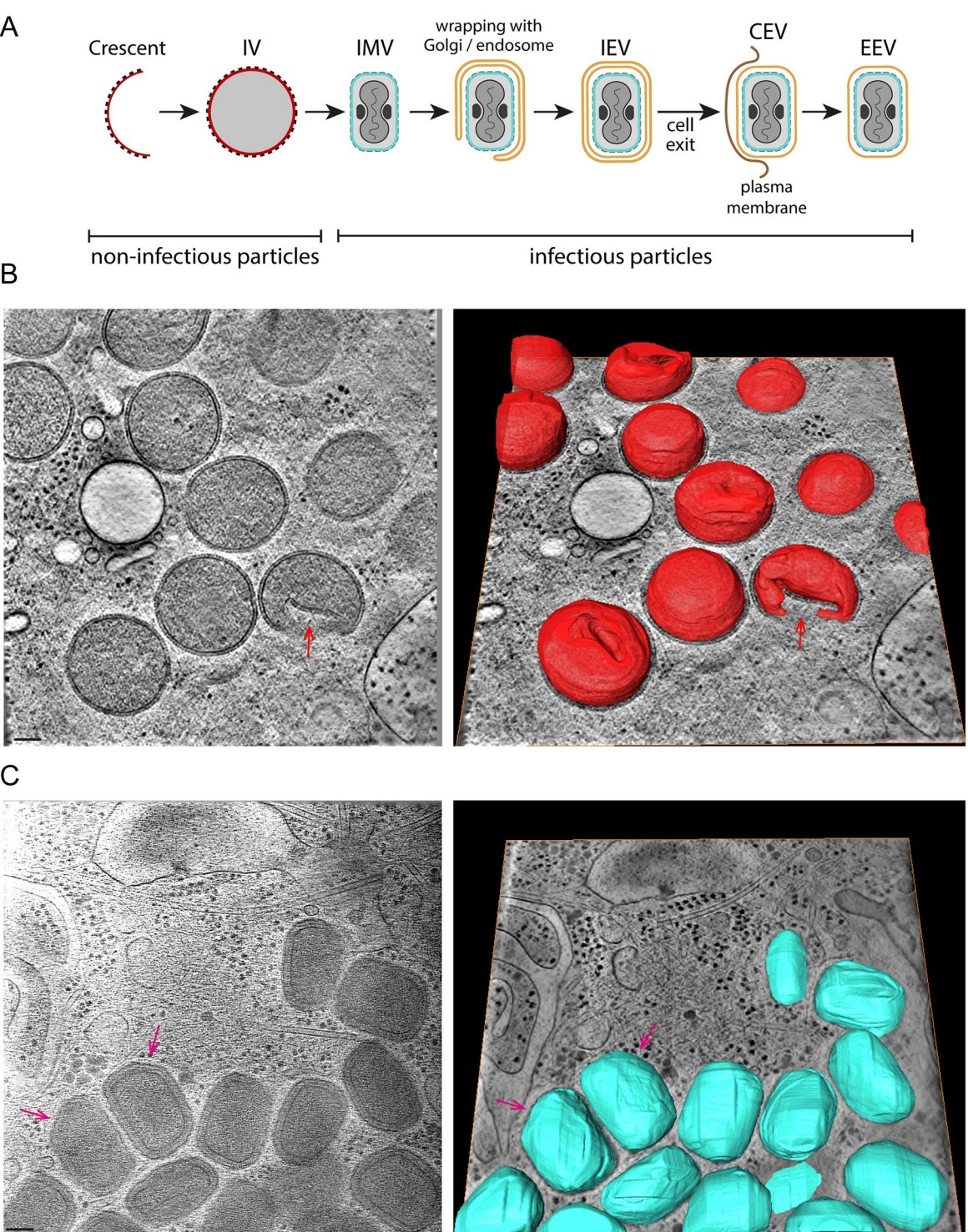

**Fig 1. Cryo-ET of vaccinia-infected cells.** (**A**) Schematic representation of vaccinia assembly and envelopment. D13-coated membrane crescents close to form a spherical IV (membrane in red), which mature into brick-shaped IMV (membrane in blue) with a viral core that presents 2 concavities, occupied by 2 lateral bodies (black). IMV can be enveloped by a Golgi cisterna or endosomal compartment (orange), resulting in the formation of triple membraned IEV, which can fuse with the plasma membrane to liberate EEV, which are called CEV if they remain attached to the cell surface. (**B**) Left: Central section of a tomogram showing fully formed IV. The red arrow points at an IV with a lateral invagination, which is detected in 8 out of the 11 IV in the tomogram. Invaginations in other IV can be seen in the 3D-segmented model of the membrane of the same tomogram (right) (see S1 Movie). The sample thickness range was 310–360 nm. (**C**) Tomogram middle view (left) and its corresponding segmentation (right) of IMV (see S2 Movie). A "cut corner" (magenta arrows) is

apparent in some IMV. Sample thickness range was 260–270 nm. Scalebars = 100 nm. CEV, cell-associated enveloped virus; cryo-ET, cryo-electron tomography; EEV, extracellular enveloped virion; IEV, intracellular enveloped virion; IMV, intracellular mature virion; IV, immature virion.

structured illumination microscopy together with deconvolution (125 nm resolution) also reveals the presence of elongated DNA structures associating with virions, identified by RFP-A3, in the viral factory (Fig 2F). Our 3D tomographic observations of assembling IV using cryo-ET were consistent with previous ultra-structural analysis of vaccinia-infected cells using electron microscopy [16–18,29,30].

## In situ structure of the hexagonal D13 lattice

Structural analysis of recombinant D13 in isolation has provided a detailed molecular understanding of the D13 trimer [19–22]. However, the macromolecular arrangements of the D13 lattice on IV in infected cells have largely been studied by deep-etch EM [17,18]. Given the clear lattices present on the IV surface in our tomograms (Fig 3A and S4 Movie), we analysed the structure of the D13 lattice in situ by STA. We obtained maps by averaging surface sectors revealing hexamers of trimer arrangements with a lattice spacing of 133 Å, in agreement with lattices described for in vitro assembled D13 and deep-etch EM honeycombs (Fig 3B). Rigid-body docking of the high-resolution structure of a D13 trimer (PDB: 7VFE) into the map shows the packing of D13 trimers is consistent with that recently reported [19].

From the hexagonal packing of D13 in the subtomogram average, we estimate that approximately 5,300 D13 trimers will cover the surface of a spherical IV. The packing density and radius of the D13 lattice on IV are compatible with an icosahedrally symmetric architecture with a triangulation number (T) in the neighbourhood of 268. We have not confirmed the presence of icosahedral symmetry, which would require pentameric arrangements of D13 trimers, such as those that have been identified in vitro [19]. Finally, we observed a gap of approximately 5.6 nm between the innermost surface of the D13 lattice and the membrane of the IV. This gap cannot be explained by any unresolved N- or C-terminal residues of D13 and must therefore be bridged by another protein such as A17 [21].

## Structure of intracellular mature virus in infected cells

The internal organisation, dimensions, and shape of IMV, which are brick-shaped membrane-bound virions, contrast dramatically with those of IV (Fig 1). Analysis of the size and shape of IMV in 3D reveals they can be approximated to a triaxial ellipsoid of $352 \times 281 \times 198$ nm with a calculated volume of $1.02 \times 10^7$ nm$^3$ (Fig 4 and S2 Table and S5 Movie). The viral membrane of IMV lacks D13 and is heavily corrugated, in contrast to the IV membrane, which is smooth (Fig 4 and S3 Fig). In addition, there is a new additional outer layer of 6.67 ± 0.11 nm on the viral membrane that was not present on IV (Figs 4, 5A and 5B). Underneath the viral membrane, there is the palisade structure consisting of a series of turret-like densities in cross-section (12.5 nm thick), which is associated with an inner wall that is 3.84 ± 0.09 nm thick (Fig 5A). Depending on the orientation of the virion, the interior organisation of the IMV appears very different. In the mid view of its two widest dimensions (352 × 281 nm), the palisade appears in contact with the viral membrane. However, in the orthogonal lateral view (352 × 198 nm) the palisade structure, which only contacts the viral membrane at the ends of the virion, presents two concavities (Fig 5B). Each cavity contains a single dense and amorphous structure, termed a lateral body. The distance between the viral membrane and the palisade varies between 9.28 ± 0.18 nm and 46.53 ± 2.72 nm, the latter being where the lateral

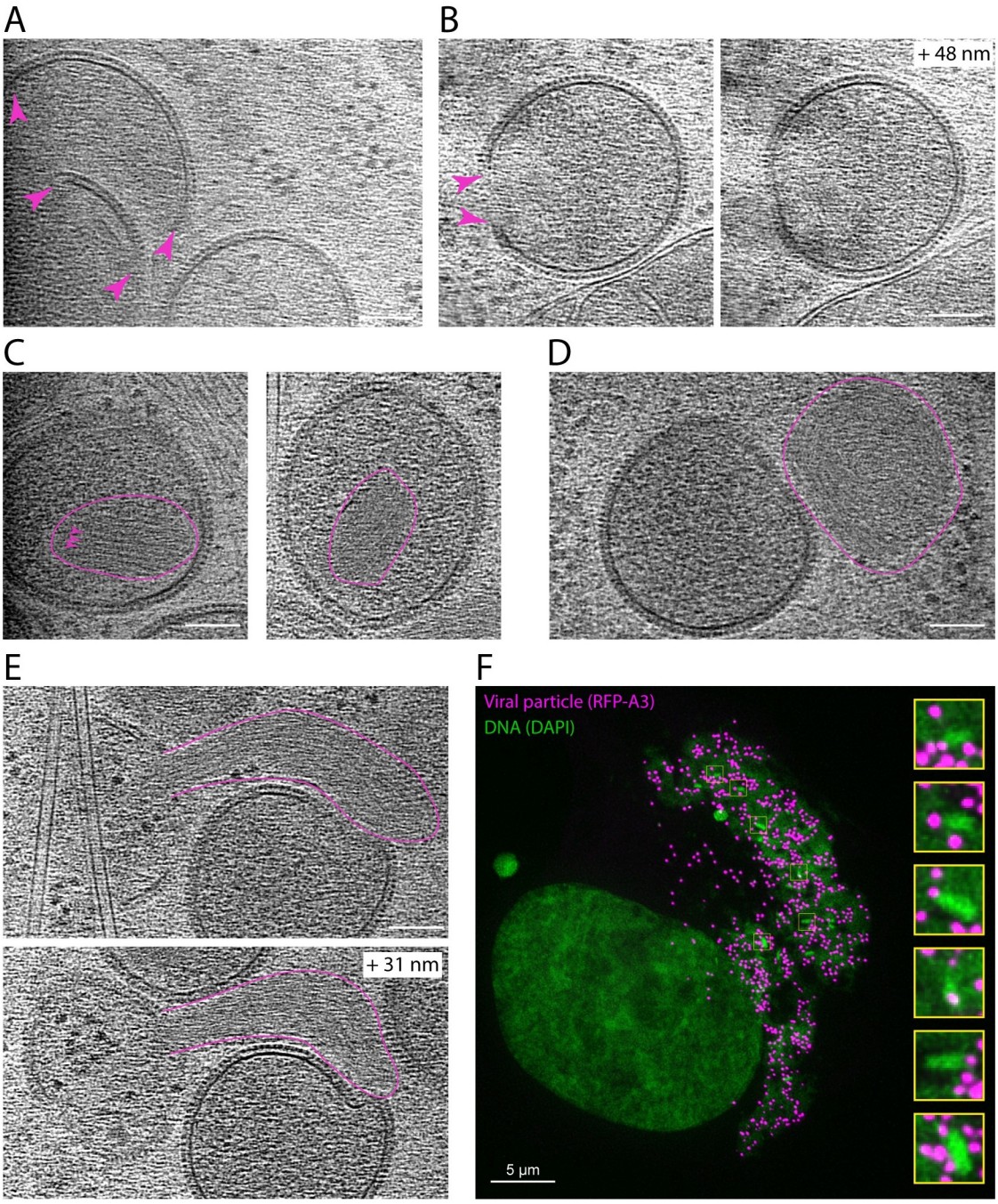

**Fig 2. The stages of IV assembly.** (**A**) Tomogram section showing D13-coated membrane crescents with open ends (magenta arrowheads). (**B**) Tomogram sections of a crescent that is almost closed: The middle view (left) reveals a membrane discontinuity (magenta arrowheads), while in a higher plane (+ 48 nm, right), the membrane is continuous (see S3 Movie). (**C**) Closed IV containing a condensed nucleoid (outlined in magenta) with repeated structural features (magenta arrowheads). (**D**) Cytoplasmic condensed nucleoid/s next to an IV. (**E**) Extended nucleoid/s associated with an open IV. (**F**) Structured illumination microscopy image of an infected cell showing condensed DNA structures (green) in association with RFP-A3 positive viral particles (magenta) located in a viral factory. Tomogram scalebars = 100 nm. IV, immature virion.

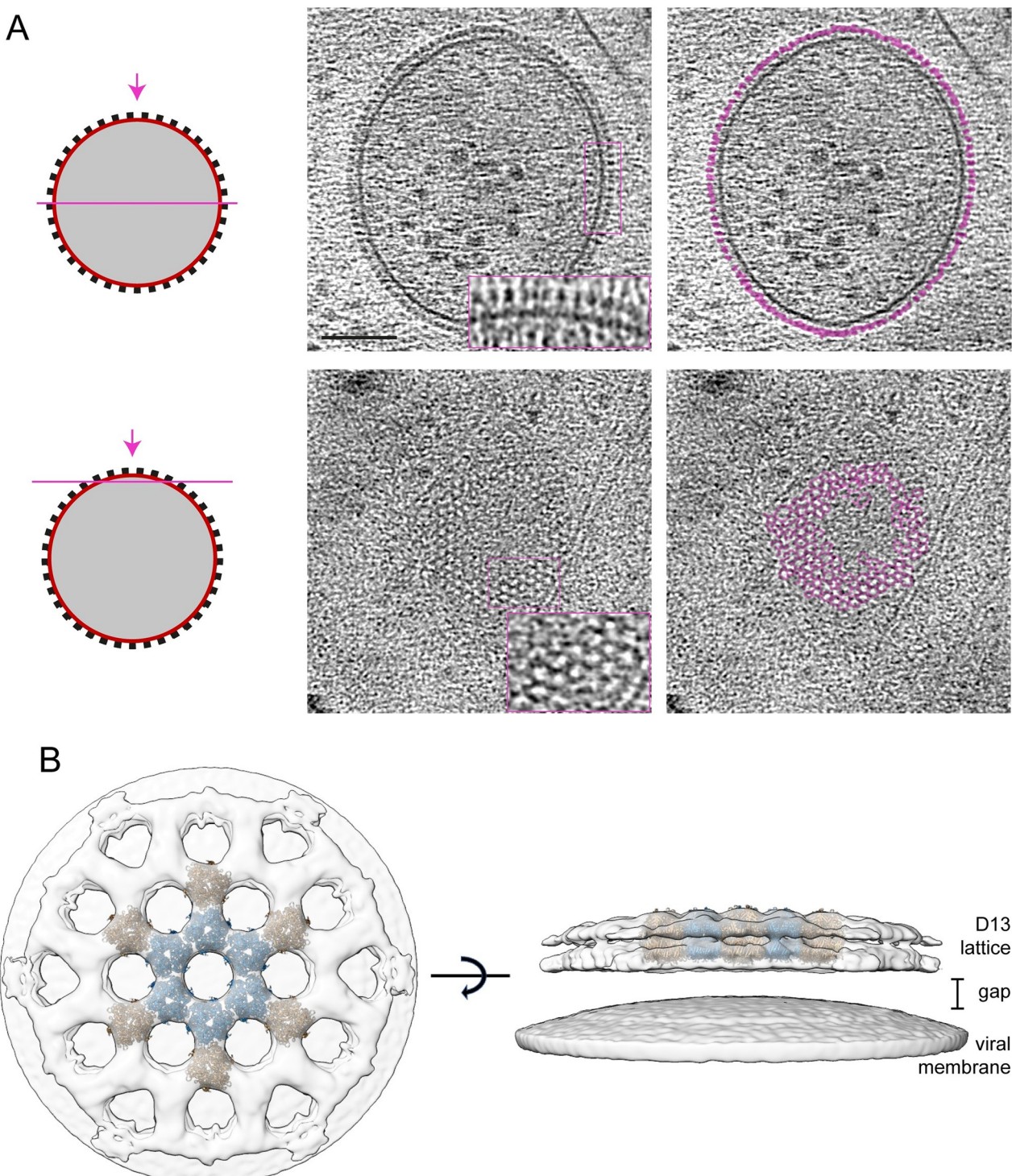

**Fig 3. In situ structure of the D13 lattice.** (**A**) Tomogram section as indicated in the schematic on the left through the middle and top of a representative IV (see S4 Movie). The densities corresponding to the D13 lattice are coloured in magenta in the right panels and show pseudohexagonal organisation. Scale bar = 100 nm. (**B**) Map based on STA of D13 and the underlying IV membrane from a top (left) and a side (right) view with fitting of coordinate models of D13 trimers (PDB 7VFE) shown in blue/brown. IV, immature virion; STA, subtomogram averaging.

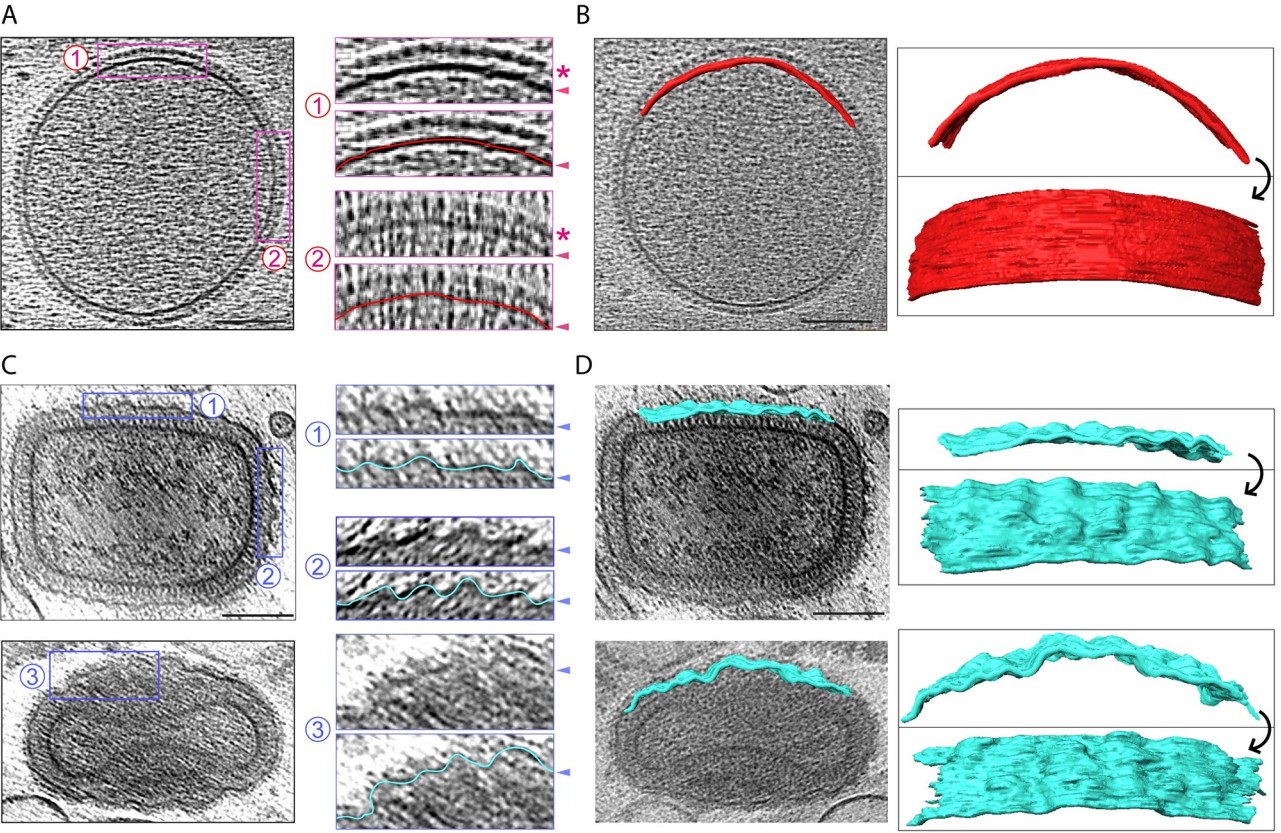

**Fig 4. Corrugation of the viral membrane during IV to IMV maturation. (A)** Representative IV showing that the viral membrane is smooth. Magnified regions (2 x) are shown to the right, with or without a line tracing the IV membrane. The D13 lattice is marked with an asterisk and arrowheads indicate the viral membrane. **(B)** A portion (680 × 65 nm) of the IV membrane was segmented and 2 segmentation views are shown illustrating its smoothness. **(C)** Top left shows a middle section of an IMV, while the bottom left image corresponds to a "side view" of another IMV. In the magnified regions, a cyan line traces the wrinkled IMV membrane (arrowheads). **(D)** A portion (570–580 × 65 nm) of each viral membrane was segmented and 2 views are shown for each virion. Scale bars = 100 nm. IMV, intracellular mature virion; IV, immature virion.

bodies are accommodated (see S2 Table for detailed measurements). 3D tomographic analyses reveal that the palisade is a semi-regular array when viewed from above (Fig 5C and 5D).

The semi-regular organisation of the palisade is most apparent in naked cores, which lack the viral membrane and are occasionally found in the cytoplasm (Fig 6A and 6B). Segmentation of our 3D tomograms reveals the palisade is a continuous structure without fenestrations that defines the boundary of the virus core, including the regions that contact the lateral bodies (Fig 5E and S6 Movie). Inside the core, there are no obvious higher order structures. There are, however, interconnected densities that vary between virions but tend to accumulate beneath the inner wall of the core (Fig 5A). These densities are especially apparent in the compressed region underneath the lateral bodies (Fig 5C). Such densities associated with the inner wall of the core potentially represent the viral genome and its associated proteins given its high contrast. Another characteristic that is evident in midplane views of approximately half of the IMV (48.9%, $n$ = 94), is that one corner of the virion (and in a few cases, 2 corners) appears as a straight or flattened "cut corner" (Fig 1C and S3 Fig). This characteristic is also apparent in naked cores (Fig 6A and S7 Movie) and is also evident in previous published electron micrographs of cores obtained from purified IMV particles [27]. This suggests the cut corner of the core is an intrinsic structural property of the palisade. In addition, these naked cores lack the two densities corresponding to lateral bodies but have associated spaghetti-like structures

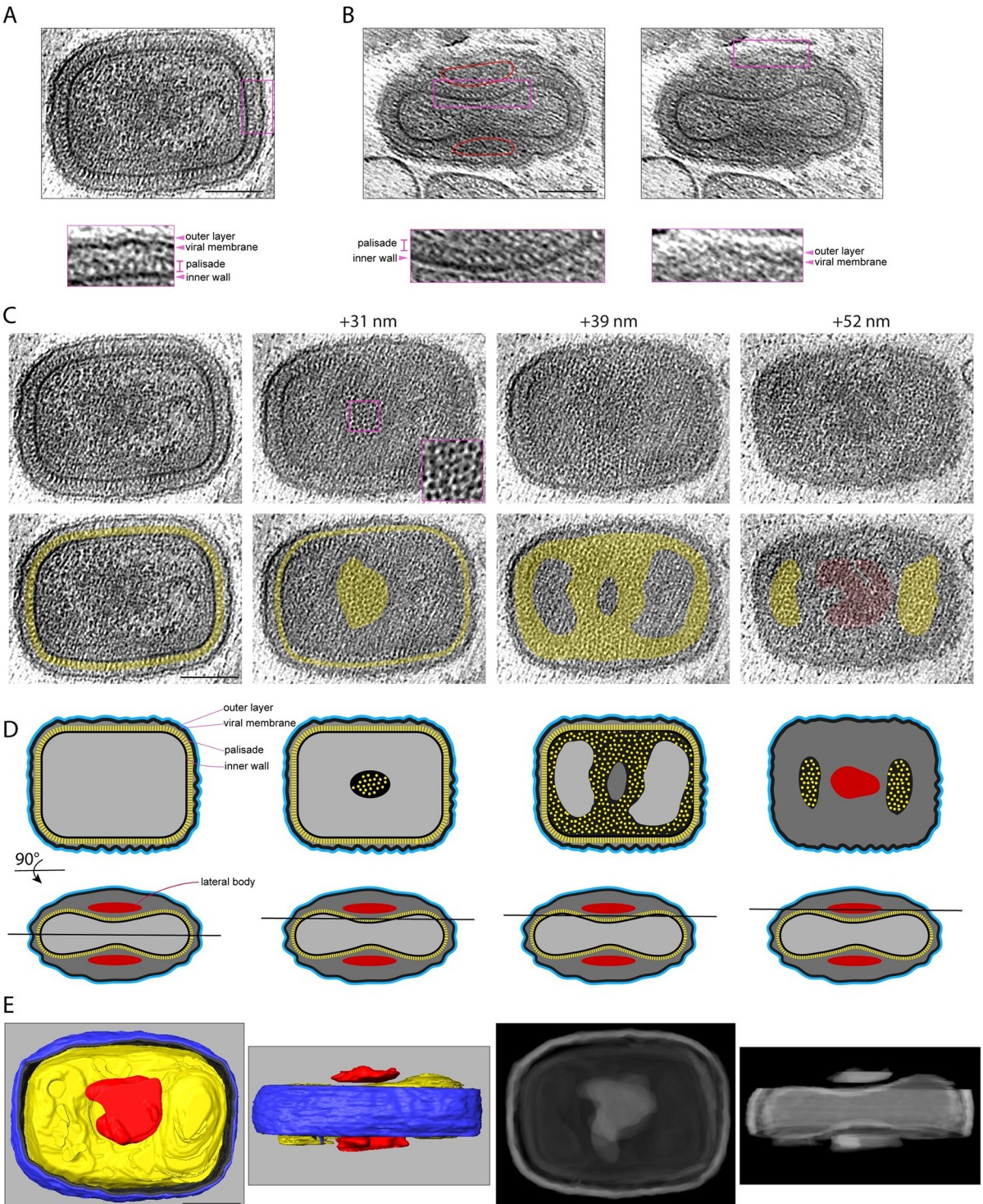

**Fig 5. Ultrastructure of IMV.** (**A**) Central tomogram section showing the broadest view of a representative IMV together with a region magnified 2.5 times to highlight the IMV layers (see S5 Movie). (**B**) Lateral view of an IMV showing the lateral bodies (outlined in red) and the two concavities of the core. The magnified regions outlined in magenta show the IMV layers. (**C**) Different sections of the IMV shown in (A) at the indicated plane positions in nm. The inset in the +31 nm view shows the surface of the palisade lattice. In the second row, yellow highlights the palisade structure, while red indicates the lateral body. (**D**) The first row shows a schematic representation of the views displayed in (C), while the second row represents the orthogonal view together with the tomogram position (black line). (**E**) Left: top and side views of a segmentation model of the IMV in (A). The outer layer (blue) and viral membrane (black) are cut away so the internal palisade (yellow) and the lateral bodies (red) can be seen (see S6 Movie). Right: The 2 concavities of the core are more apparent in a Simulated Digitally Reconstructed Radiograph (DDR Rendering) of the same segmentation model. Scale bars = 100 nm. IMV, intracellular mature virion.

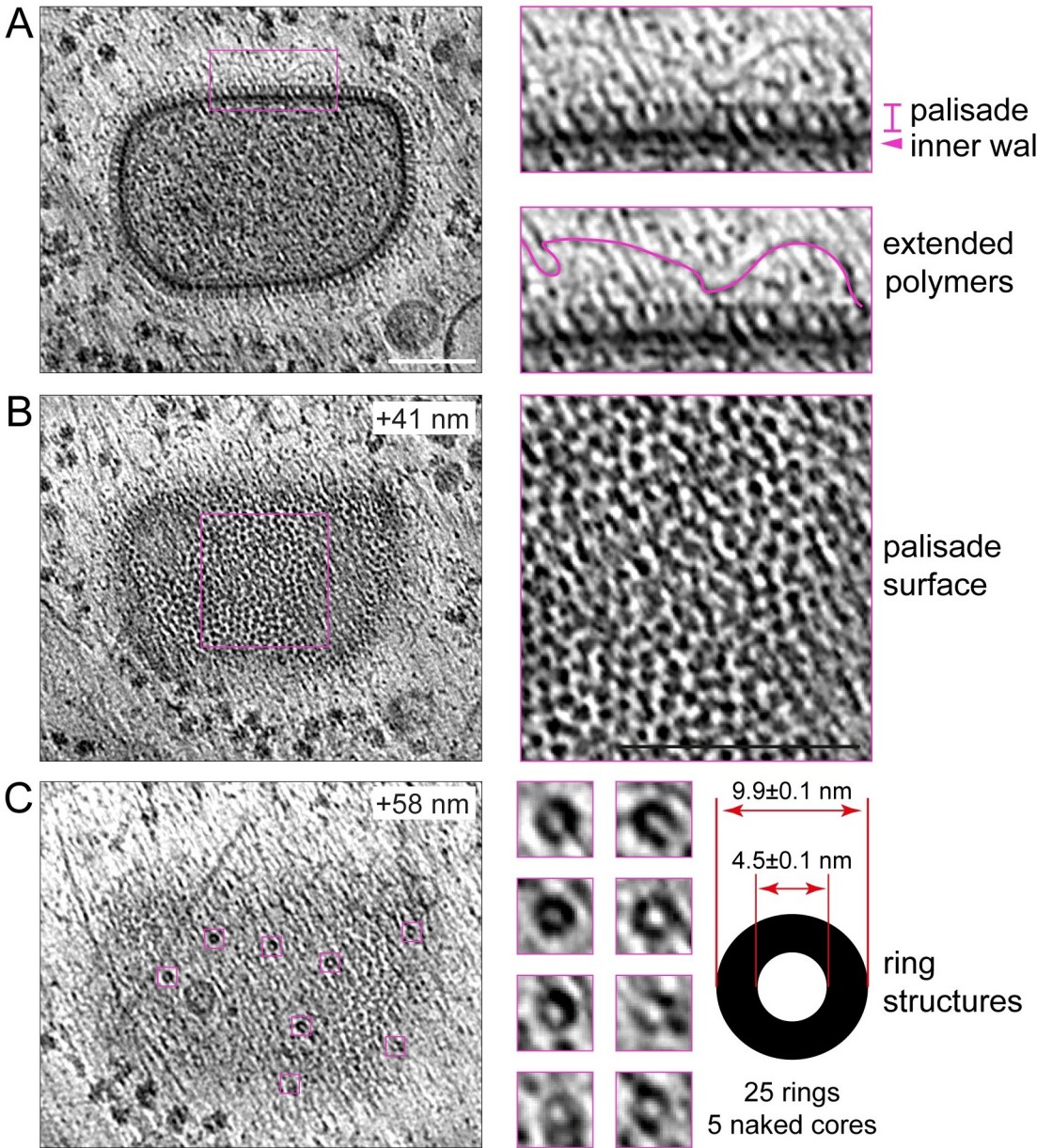

**Fig 6. Ultrastructure of the naked viral core.** (**A**) Middle section of a naked viral core together with a region magnified 3 times to highlight the palisade and inner wall as well as the polymers that surround the core. (**B**) A higher tomogram section (+41 nm) of the same naked viral core highlighting the pseudohexagonal lattice of the palisade. (**C**) At an even higher section (+58 nm), ring structures associated with the palisade are evident. The dimensions of the rings ($n$ = 25) from 5 naked cores are indicated together with the standard error of the mean. Scale bars = 100 nm.

contacting the palisade surface (Fig 6A). These flexible polymers, which are 2.6 nm in diameter, form an exclusion zone of approximately 40 nm around all 5 naked cores we observed (Fig 6A). Furthermore, it was noticeable that naked cores have randomly distributed ring-like structures on their surface that were not observed in IMV. These rings seem to protrude 10 to 20 nm from the palisade surface and had inner and outer diameters of 4.5 ± 0.1 nm and 9.9 ± 0.1 nm, respectively (Fig 6C).

### The corrugated membrane and hexagonal core lattice of vaccinia virions

During viral egress, some IMV become triple-membraned IEV after envelopment by a Golgi cisterna or endosomal compartment (Fig 1A). Subsequent fusion of the IEV with the plasma membrane releases double-membraned EEV that are known as CEV if they remain attached to the outside of the cell [7]. Our tomograms reveal that the additional membranes acquired by envelopment are smooth and not corrugated as observed for the IMV membrane (Fig 7). Both leaflets of the smooth CEV membrane are discernible in un-binned tomogram sections (S4 Fig). In the case of IEV, the outermost membrane is also not always in close contact with the underlying membrane (see S2 Table for virion measurements). The dimensions and structure of the inner IMV and corrugated membrane are also unaffected by envelopment (Fig 7B). Notably, the palisade fully coats the viral core in all infectious virions and its organisation appears unaltered (Figs 7A and 8A). To study the architecture of the palisade, we performed STA separately using IMV, IEV, and CEV/EEV particles. Maps obtained from the different virion types, all displayed the same organisation and lattice parameters (a = b = 89 ± 2 Å, θ = 120°) (S5 Fig). A new combined map obtained by averaging all particle types together, reveals that the palisade is composed of trimeric pillars with projecting lobes that interact with neighbouring pillars with local hexagonal symmetry (Fig 8B). These pillars are embedded in an unfeatured inner wall. While further details are required to understand its molecular composition, this arrangement appears to be flexible enough to assemble a continuous biconcave capsid structure.

### Maturation of IV to IMV

Radical changes in the organisation, dimensions, and shape of IV result in the formation of IMV (Fig 9). Based on our tomograms, IV have an average diameter of 351.89 ± 2.88 nm and a volume of $2.28 \times 10^7$ nm$^3$ for completely spherical particles (Fig 9A). In contrast, "brick-shaped" IMV have a volume of $1.02 \times 10^7$ nm$^3$ based on dimensions of 352 × 281 × 198 nm (Fig 9A). This reduction in volume is accompanied by a dramatic corrugation of the viral membrane, together with the loss of the D13 lattice (Fig 4). To better characterise membrane corrugation, we measured the middle-plane perimeter of the viral membrane in IV and IMV, following the wrinkles of the viral membrane to obtain its contour length. We found that the membrane contour is virtually identical in IV and IMV, as well as in the equivalent innermost membrane of EEV/CEV (approximately 1,100 nm, Fig 9B). This suggests the viral membrane folds during maturation, which would explain the reduction in volume without any detectable loss of membrane surface. In addition to reducing their volume by approximately 50%, IV also change their shape when they mature into IMV, becoming a triaxial ellipsoid. Strikingly, the longest IMV dimension matches the diameter of IV (Fig 9A and 9B), suggesting that the major axis of IMV is determined by the IV diameter. In our tomograms, we also found particles that may represent intermediates and/or defective examples of IV maturation. These include examples where the palisade is fully formed but the viral membrane, which either lacks or is partially coated with D13, is not associated with the viral core (Fig 9C). It is also interesting that we did not observe any virions with partially formed cores suggesting that palisade formation is likely to be rapid or occur en bloc.

## Discussion

Previous ultra-structural analyses of vaccinia-infected cells by electron microscopy over many decades have provided important insights into the assembly of vaccinia virions. These studies, however, have largely been conducted on fixed sections of infected cells that can have fixation, processing, and staining artifacts. Cryo-EM studies, analysing viral entry or purified virions,

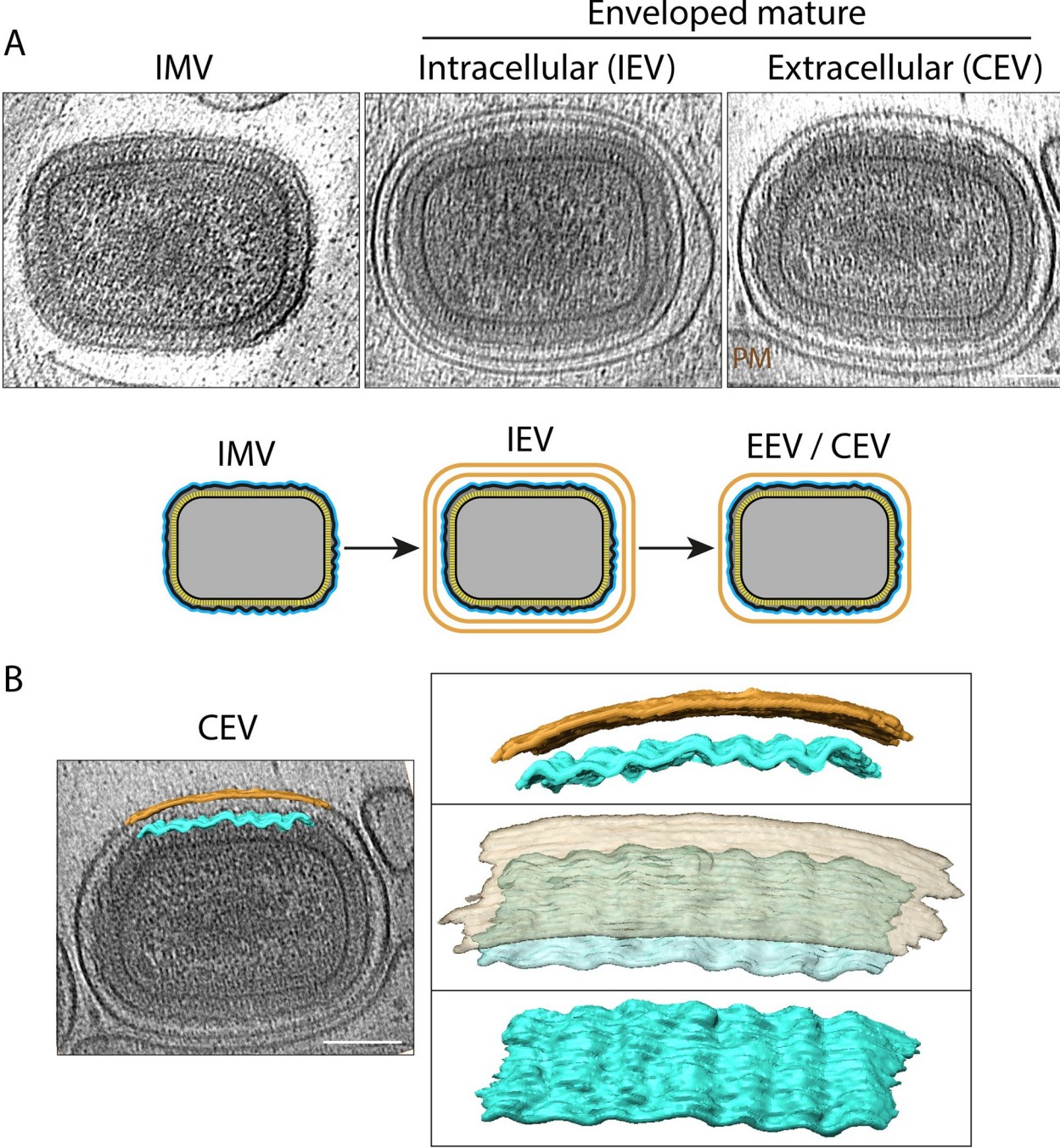

**Fig 7. The invariant architecture of the corrugated IMV membrane.** (**A**) Central plane of tomograms showing that the additional membranes of IEV and CEV (associated with the PM) do not alter the ultrastructure of the internal IMV. A schematic representation illustrating the number of membranes of IMV, IEV, and EEV/CEV is shown beneath the tomograms. (**B**) Segmented membranes of the CEV shown in (A), highlighting the corrugated character of the inner membrane, which is in contrast to the smooth outer membrane of the CEV. The images on the right show views of the segmented portions of the inner (cyan, 450 × 65 nm) and outer (orange, 520 × 65 nm) membranes. Scale bars = 100 nm. CEV, cell-associated enveloped virus; EEV, extracellular enveloped virion; IEV, intracellular enveloped virion; IMV, intracellular mature virion; PM, plasma membrane.

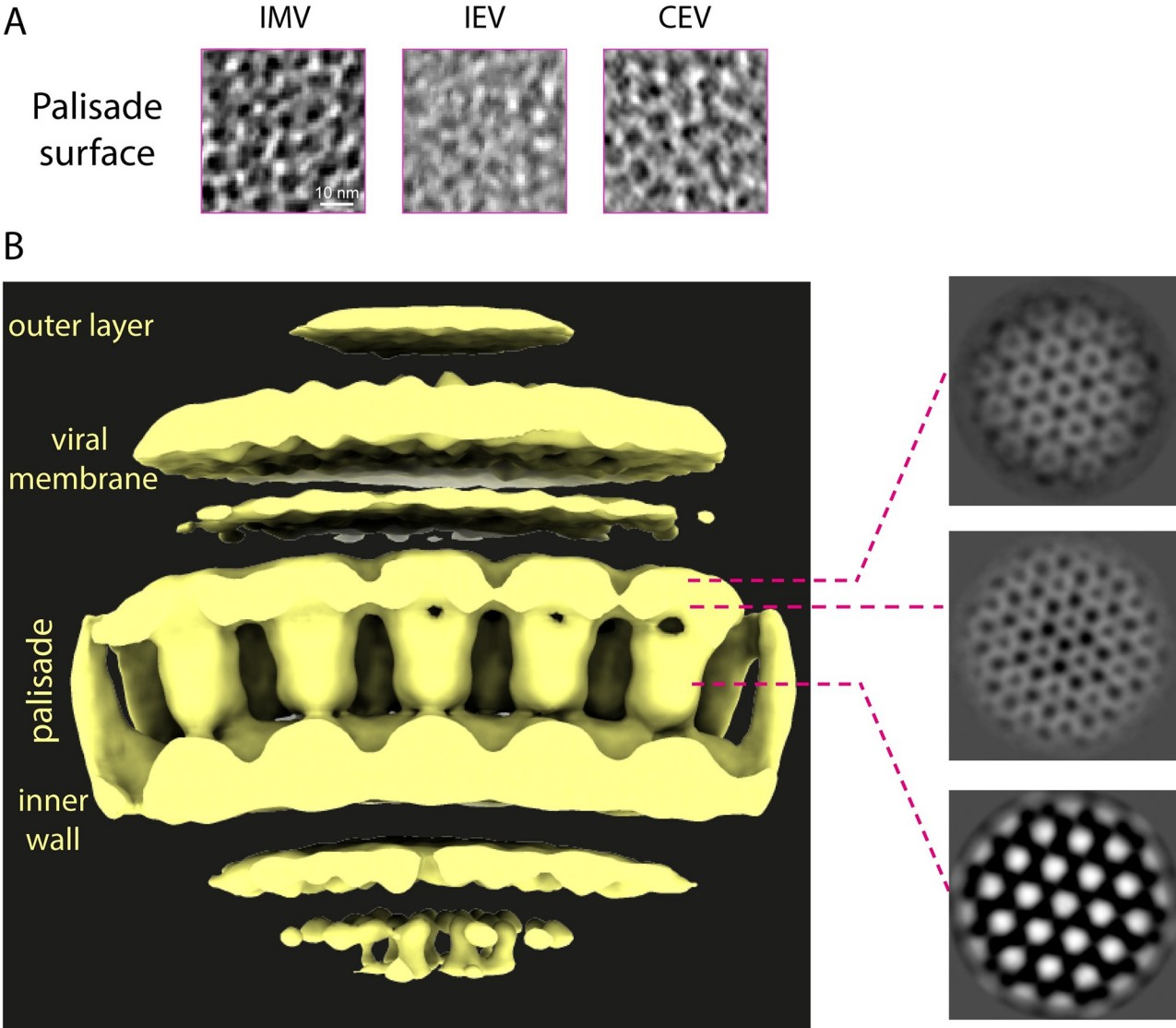

**Fig 8. The palisade is a pseudohexagonal lattice composed of trimeric proteins.** (A) Sections through the palisade layer corresponding to the virions shown in Fig 7A. Scale bar = 10 nm. (**B**) Map derived from STA showing the inner wall, palisade layer, viral membrane, and outer layer in surface representation (left) and corresponding sections of the map (right, greyscale). CEV, cell-associated enveloped virus; IEV, intracellular enveloped virion; IMV, intracellular mature virion; STA, subtomogram averaging.

have demonstrated the potential of using cryo-EM to study vaccinia structure [26,27,31,32]. Here, we apply cryo-ET to image the thin edge of plunge-frozen, vaccinia virus-infected cells, revealing virus architecture in situ. While there is confirmation of past work, important new features emerged from our study. We found that during maturation to IMV, spherical IV lose their D13-coat and reduce their volume by approximately 50%. During this process, the outer viral membrane becomes corrugated and contacts the capsid-like palisade layer and the lateral bodies. The palisade, which remains unaltered during subsequent virion morphogenesis, is a continuous regular lattice with pseudohexagonal symmetry that defines the viral core boundaries. Our observations suggest that these two lattices, D13 and the palisade, drive vaccinia assembly and maturation.

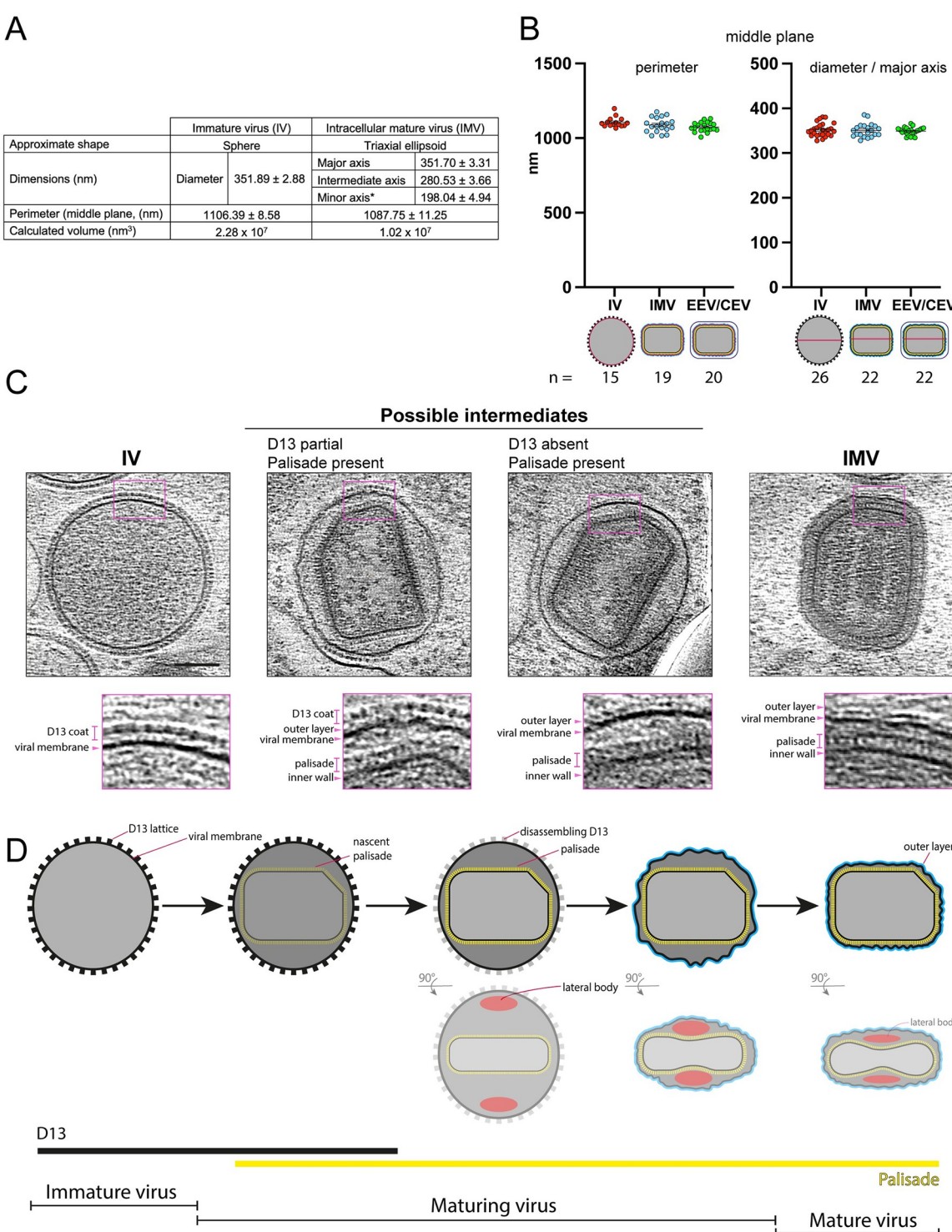

**Fig 9. Maturation of vaccinia virus: From IV to IMV.** (**A**) Middle view dimensions of IV (diameter, *n* = 26; perimeter, *n* = 15) and IMV (major axis, *n* = 22; intermediate axis, *n* = 19; minor axis, *n* = 10; perimeter, *n* = 19) tomograms. * Indicates that the IMV minor axis was calculated from the side views of 3 IMV and 7 IMV inside EEV (see Methods). For volume calculation, IV and IMV were assumed to be spheres and triaxial ellipsoids, respectively. S3 Table shows the individual numerical data. (**B**) Plots showing the individual values for perimeter and diameter/major axis (magenta lines in schematics below graph) of the IV, IMV, and EEV/CEV. IV and IMV measurements were used to calculate the averages shown in (A). The averages of the EEV inner-most membrane perimeter and major axis are 1,074.46 nm ± 7.09 nm and 348.91 ± 1.87 nm, respectively. (**C**) IV and IMV along with possible intermediates together with magnified regions (2.5 times) highlighting the different layers of each particle. Scale bar = 100 nm. (**D**) Vaccinia

maturation model: the palisade lattice (yellow) forms inside D13-coated IV, its maximum length being determined by the IV diameter. Following disassembly of the D13 lattice, the viral membrane acquires the shape of the palisade and becomes wrinkled. During this process, the presence of two lateral bodies (red) results in the deformation of the core, which adopts its characteristic biconcave shape. All errors in (A) and (B) are given as ± standard error of the mean. EEV, extracellular enveloped virion; IMV, intracellular mature virion; IV, immature virion.

We found that the D13 lattices coating IV have a similar architecture to the D13 lattices formed in vitro [19]. Furthermore, Hyun and colleagues produced D13 spherical IV-like particles with a similar diameter to the D13-coated IV we imaged in situ. This strongly suggests that the D13 lattice, which initially curves open membranes and eventually forms closed particles, determines both the size and shape of the IV. In addition, our images and STA maps show that a 5.6-nm gap separates the D13 lattice from the IV membrane. This gap is most likely occupied by A17, which tethers the D13 lattice to the membrane [21,33,34] but whose flexibility may preclude its detection on the IV membrane by cryo-ET. The lack of direct contact of the D13 protein lattice with the viral membrane is reminiscent of clathrin and other vesicle-forming coats in eukaryotic cells where the structural lattice does not directly interact with the membrane to be deformed [35,36]. Instead, an adaptor module mediates the lattice-membrane association. A mechanism by which coat complexes promote membrane curvature is by clustering the adaptor module on the membrane to be deformed [37,38]. Therefore, D13-mediated clustering of A17 could promote membrane deformation during IV formation. In addition, A17 can also contribute to membrane bending independently of D13, since it can deform membranes in vitro and when expressed in non-infected cells [39].

Despite their relatively constant diameter, a proportion of D13-coated IV have an elongated membrane invagination (Fig 1A). This feature has largely gone unnoticed in previous EM studies imaging thin sections, although IV membrane deformations are apparent in some studies [28,40–43]. While the cause of this membrane buckling remains to be established, it may relate to the thinness of the cell periphery or a consequence of forces on the membrane during IV assembly or DNA insertion into assembling IV [42]. As in other studies, we observe the nucleoid in the IV interior or associated with either open or closed spheres, as previously suggested for DNA entry events [28,30]. Any possible relationship between the IV invagination and the site of DNA entry and IV sealing will require future analysis.

The palisade, which is largely composed of p4A and A4 [31,44,45], consists of a trimeric assembly in a pseudohexagonal lattice arrangement. While high-resolution data is required to interpret the structure of the palisade, the estimated mass of a single palisade trimer based on contouring of our STA map (around 240 to 360 kDa) is consistent with the mass of a p4a/A4 trimer of heterodimers (305.8 kDa). Preliminary fitting of 3 p4a/A4 heterodimers, derived from the structural prediction of Alphafold into our STA map is consistent with the overall size and shape of the palisade trimer. Based on the uniformity of the IMV and the relative measurements we obtained of IV and IMV, we conclude that this new viral lattice, which fully covers the core and defines its boundary, dictates the dimensions and shape of IMV. The viral membrane, no longer covered by D13, would wrinkle and adopt the shape and dimensions of the newly formed palisade. This process may also be responsible for the compression of the lateral bodies onto the virus core, causing the biconcave deformation of the latter. The maturation model that emerges from our work provides a simple way by which a membrane-bound particle adapts to a new internal lattice and changes its shape and dimensions to become more compact without membrane removal (Fig 9D). Moreover, our data suggest the palisade length is dictated by the diameter of the IV, which in turn is established by D13. The palisade would grow inside the D13-coated IV until reaching its maximum length, that is the IV diameter. In other words, the D13 lattice limits the longest dimension of the palisade lattice. The size restriction of a growing

viral core by its surrounding viral membrane has also been proposed for the assembly of the HIV core [46,47]. Our observation of a viral particle partly coated by D13 with an assembled palisade suggests the palisade forms before D13 disassembles and also implies that complete removal of D13 is not required for palisade formation (Fig 9C). Moreover, our observations suggest that palisade assembly and D13 removal are coordinated and rapid.

In the IMV, the lateral bodies define a lateral domain, as opposed to the virion tips, which are specified by the long axis of the viral core. The lateral bodies prevent direct interactions between the palisade and the wrinkled viral membrane, which might contribute to the differences between the lateral and the tip domains. Notably, the virion tips are the site of polarisation of the set of viral membrane proteins that form the entry fusion complex (EFC), which is essential for virus infectivity [48–50]. How and when EFC polarity is established is unknown. An appealing possibility is that virion tips are stochastically defined by the growth of the palisade long axis during assembly and the EFC and other viral factors polarise at the virion tips after palisade formation by their exclusion from the lateral domains. In such a scenario, palisade formation would drive virion polarity.

The EFC mediates fusion of the IMV membrane with the plasma membrane or an endocytic compartment during entry, which releases a naked core into the cytoplasm [51–55]. Subsequently, early proteins are produced by early RNA transcripts released from naked cores, which are essential to liberate the DNA genome into the cytosol to initiate replication [1]. Our cryo-ET data of naked cores in infected cells indicate the palisade structure does not require surrounding membranes for stability. Furthermore, the loss of lateral bodies from naked cores suggests their association with the palisade depends on the IMV membrane. In addition, we observed flexible 2.6-nm thick polymers fully surrounding naked cores that might correspond to either RNA or the viral genome that is released from the naked core. Finally, we detected ring structures protruding from the palisade surface of naked cores. Analysis of these rings reveals they are compatible in dimension with the hexameric rings of the viral D5 primase/helicase, which is essential for vaccinia genome release [56,57]. A similar pore-like structure has been previously described on the surface of vaccinia cores generated by treating purified IMV with NP40 and DTT [58]. In addition, pores in the palisade itself, and not protruding rings, have also been reported in intact purified IMV [26]. Further work is needed to determine the identity and relationship between these pore-like structures, as D5 only associates with naked cores after entry and not newly assembled IMV [56].

A common feature of many viral families is the use of a 3D lattice, or capsid, which determines the structure and dimensions of the virion. What distinguishes vaccinia from other viruses is that an initial lattice, D13, is replaced by a second lattice, the palisade, during virion assembly. This contrasts African swine fever virus (ASFV), which retains its D13-like lattice, in addition to an inner capsid in mature virions [59]. Based on the structural similarities between D13 and the structural components of other viruses with a pseudo-hexameric structure formed by a trimeric protein containing concatenated beta-barrels, vaccinia has been included in the PRD1/adenovirus lineage [60]. However, the viral proteins that form the palisade, which is present in the mature particles, may be just as important as D13 in structural phylogenetic comparisons.

The palisade is a regular capsid-like structure determining the morphology of all 4 forms of infectious vaccinia virions (IMV, IEV, EEV, and CEV). This palisade is surrounded by a heavily corrugated membrane. Our study shows that the palisade and the corrugated viral membrane are invariant defining features of all types of infectious virions. The additional membranes acquired by envelopment are different to the IMV viral membrane as they are not corrugated or always in close contact with the IMV surface (Fig 7). Moreover, the acquisition of these membranes is not associated with major changes in the virion structure or palisade architecture (Figs 7 and 8). This suggests that these additional membranes are used to facilitate

viral egress before cell lysis. In fact, these membranes and associated proteins drive IEV transport on microtubules to the cell periphery, IEV fusion with the plasma membrane and subsequent actin-based transport of CEV [7–11,13]. In contrast, the IMV membrane serves to contain the viral core and lateral bodies and organises the essential components required for entry and, also, envelopment during viral morphogenesis. Based on the strong protein conservation between orthopoxviruses, we believe that the virion assembly of monkeypox and variola major are likely to be identical to that of vaccinia virus we have described here.

## Methods

### Recombinant viruses

To facilitate cryo-ET observations, we produced recombinant Western Reserve vaccinia strains lacking F11 to prevent cells from rounding-up early during infection [61]. In addition, the A36-YdF mutation was used to abolish actin tail formation beneath CEV [11,13,62]. The F11 gene was deleted in A36 YdF [8] or A36 YdF ΔNPF1-3 [63] backgrounds using the same targeting strategy as previously described [64]. Fluorescence was used as a selectable marker to isolate recombinant ΔF11-mCherry viruses by successive rounds of plaque purification in BS-C-1 cells. Correct gene replacement was confirmed by PCR, sequencing of the F11 locus, and western blot analysis.

### Cell growth, vaccinia infection, and vitrification

HeLa cells were maintained in complete MEM (supplemented with 10% fetal bovine serum, 100 ug/ml streptomycin, and 100 U/ml penicillin) at 37 ˚C with 5% $CO_2$. Cells were washed with PBS, treated with trypsin, seeded on glow discharged (40 s at 45 mA) Quantifoil R3.5/1 gold grids of 200 mesh and placed in wells of 6-well plates with complete MEM. After overnight growth, the cells were infected with A36-YdF ΔF11 or A36-YdF ΔNPF1-3 ΔF11 vaccinia strains in serum-free MEM at a multiplicity of infection of 2. After 1 h, the medium was replaced with complete MEM. At 8 h post infection, grids were washed once with PBS and excess PBS was removed with a Whatman paper before blotting and vitrification using the Vitrobot Mark IV System, which was set to 95% relative humidity at 22˚C. Colloidal gold particles (10-nm diameter) were pipetted onto grids before blotting on both sides of the grid, which was performed for 14 s with a relative force of −10.

### Cryo-electron tomography, image processing, and analyses

For data collection, we chose magnification and defocus conditions to achieve a wide field of view to capture all vaccinia assembly and envelopment stages while retaining sufficient detail of viral structures. Grids were first screened on a Talos Arctica TEM (Thermo Fisher) to select edges of cells with an abundance of viral particles that were thin enough for cryo-ET. Selected grid regions were registered before transferring to a Titan Krios (Thermo Fisher), where the mapped grid selections were re-imaged. The Titan Krios was fitted with a K2 Summit direct detector (Gatan) operated in electron counting mode. A Gatan GIF energy filter was used in zero-loss mode with a 20 eV slit width. Dose-symmetric tilt series were collected from −57˚ to +57˚ at a 3˚ increment, a pixel size of 4.31 Å and a defocus of −8 μm using Tomography 5.7 software (Thermo Fisher) (S4 Table). Four movie frames were collected per tilt with a dose of 1.7 e/Å$^2$ per tilt, giving a cumulative dose of 66.3 e/Å$^2$ per tilt series. Movie frames were aligned using alignframes from IMOD [65]. Tilt series were aligned using gold fiducials or patch tracking in IMOD. Contrast transfer function (CTF) correction and tomogram reconstruction were also implemented in IMOD [65,66]. Alternatively, tomograms were CTF-corrected and

reconstructed using novaCTF [67]. In both cases, a SIRT-like filter equivalent to 5 iterations was applied. Segmentations were manually performed (AMIRA, Thermo Fisher), in some cases on tomograms that were denoised using ISONet with CTF deconvolution and missing-wedge filling from deep learning [68]. ISONet denoising was not used during STA.

The IV diameter, the axes of IMV and their perimeters, and the major axes of IEV and EEV were measured in 3dmod from IMOD using a middle plane of each viral particle. For perimeters, the viral membranes were manually traced to measure their total contour length. As reported previously [23], virions with the side view perpendicular to the electron beam, in which 2 concavities of the core are visible, were much less frequent than other views. Because of this, for the calculation of the minor axis of IMV, we measured the minor axis of both IMV and IMV inside EEV. This is also the reason why we did not find enough side views of IEV to directly measure their minor axis (S2 Table). To estimate the IEV minor axis, we measured the thickness that the additional IEV membranes add and combined it to the calculated IMV minor axis. For volumes calculation, we assumed IV are perfectly spherical and IMV, IEV, and EEV triaxial elipsoids.

## Subtomogram averaging

Tomograms containing IV or IMV were selected for STA. D13 and palisade particles in immature and mature virions, respectively, were picked from the bin4 SIRT-filtered tomograms using oversampled surface models in Dynamo [69], then imported to Relion 3.1 [70] for subsequent STA. Unless otherwise specified, local angular searches about the initial orientations generated in Dynamo were used throughout data processing. For particle extraction, only unbinned weighted back projections reconstructions from novaCTF were used. Lattice measurements were carried out in real space using ChimeraX [71]. D13 particles were extracted in 96-pixel boxes at the bin2 pixel size of 4.31 Å/pixel. An initial reference was generated using reference-free 3D classification without applied symmetry, with the resulting map clearly showing the honeycomb lattice of D13 trimers. Initial classification and refinement was carried out in both C1 and C6 symmetries to verify the symmetry of the D13 hexamer-of-trimers. After initial classifications and removing of overlapping particles with a distance cutoff of 50Å, particles were re-extracted without binning (4.31 Å/pixel) in 192 pixel boxes. Final refinement in C6 symmetry reached 19 Å resolution according to the FSC = 0.143 criterion.

Palisade particles were initially extracted at binning level 2 (8.62 Å/pixel) in 96-pixel boxes. Initial references were generated using independent reference-free 3D classifications for each virion type. This clearly showed the palisade lattice in every case, and the particles for each virion type were from then on processed independently, but in an identical manner. For each virion type, alternating rounds of 3D classification and refinement were used to centre the palisade particles at 2-fold binning. After convergence at 2-fold binning, particles were re-extracted without binning (4.31 Å/pixel) in 128-pixel boxes. Overlapping particles were removed with a distance cutoff of 40 Å before further refinements were carried out without binning. As all refinements converged on maps with identical lattices, particles from all virion types except for the naked cores were then combined for further refinements. All refinements were carried out without application of symmetry up to this point. After the C1 maps were merged and the symmetry of the lattice was apparent (S6 Fig), further refinement runs were carried out with C3 symmetry applied. The final resolutions of the palisade maps ranged from 20 to 30 Å, according to the FSC = 0.143 criterion.

## Immunofluorescence imaging

HeLa cells on fibronectin-coated coverslips were infected with an RFP-A3 vaccinia strain [72] for 8 h and fixed with 4% paraformaldehyde in PBS for 10 min, then permeabilised with 0.1%

Triton X-100 in PBS for 5 min and incubated in blocking buffer (10 mM MES (pH 6.1), 150 mM NaCl, 5 mM EGTA, 5 mM MgCl$_2$, and 5 mM glucose) containing 2% (v/v) fetal calf serum and 1% (w/v) BSA for 30 min prior to addition of 4′,6-diamidino-2-phenylindole (DAPI) for 5 min, before mounting the coverslips using Mowiol. Coverslips were imaged using Structured Illumination Microscopy (VT-iSIM) on an Olympus iX83 Microscope with Olympus 150x/1.45 NA X-Line Apochromatic Objective Lens, dual Photometrics BSI-Express sCMOS cameras, and CoolLED pE-300 Light Source (Visitech) and was controlled using Micro-Manager 2.0.0. Image stacks of 10 to 15 z-slices with 0.1 μm steps were acquired and deconvolved using the express deconvolution setting on Huygens Software (Scientific Volume Imaging). Measurement of the point spread function of sub-diffraction beads (100 nm) confirmed that the XY resolution of the imaging system is 125 nm.

## Supporting information

**S1 Table. Sample thickness in the main figure tomograms.** Sample thickness was measured manually in the regions corresponding to the main figure tomograms using IMOD. In many cases, the presence of viral particles locally increases the thickness of the cell. For the larger regions in Fig 1B and 1C, a thickness range is provided in the corresponding figure legend (see Fig 1B and 1C).
(DOCX)

**S2 Table. Measurements and calculations of virion dimensions.** (**A**) The middle-view perimeters, IV diameter, the major axis of IMV and EEV/CEV are shown, together with the standard error of the mean and the number of viral particles measured. The intermediate and minor axes of IMV are also shown, as well as the IEV axes. Finally, the calculated volume of IV, IMV, and IEV are provided. See Methods for details of dimension calculations. (**B**) For the thickness of the inner wall and the outer layer, 5 different IMV were measured by tracing 5 different lines for each IMV, which correspond to the 5 measurements shown. To estimate the distance between the palisade and the viral membrane (with no lateral bodies), 5 IMV were used. For the distance between the palisade and the plasma membrane through lateral bodies 1 IMV and 4 EEV were included in the quantification. All values correspond to nm.
(DOCX)

**S3 Table. Individual values for virus dimensions.**
(XLSX)

**S4 Table. Cryo-ET data collection and STA processing.**
(DOCX)

**S1 Fig. Low-magnification images of vaccinia infected HeLa cells on grids.** (**A**) The pink circle marks the position where the IV in Fig 1B were imaged. The image on the right is a magnification of the circled area and its surroundings. (**B**) A cryo-EM image showing IV (magenta arrow) and IMV (green arrow) visualised at low magnification in cells over holes in the carbon support film. Scale bars = 10 (left) and 1 μm (right) (A) and 1 μm (B).
(TIF)

**S2 Fig. Tomographic gallery of IV with nucleoids.** Tomographic sections showing IV with an internal condensed nucleoid outlined in magenta in the bottom row. Scale bar = 100 nm.
(TIF)

**S3 Fig. Examples of IMV with and without cut corners.** (**A**) Gallery of tomographic sections of IMV with no cut corner, with magnified regions (2×) to highlight the wrinkled viral

membrane. (**B**) IMV with cut or flattened corner/s (black arrowheads). The magnified region of the first IMV shows the corrugation of the viral membrane, as in (A). Scale bars = 100 nm.
(TIF)

**S4 Fig. Tomogram section of CEV particles.** The CEV outer membrane in the unbinned tomogram appears as 2 parallel densities, consistent with the bilayer organisation of membranes. The CEV on the right corresponds to the one shown on Fig 7. The red square corresponds to the enlarged image in which arrowheads point to the 2 leaflets on the CEV outer membranes. Top scale bar = 100 nm. Bottom scale bar = 50 nm.
(TIF)

**S5 Fig. Palisade maps from IMV, IEV, and EEV/CEV.** (**A**) Maps derived from subtomogram averaging IMV, IEV, or EEV/CEV. A surface (top) and a cut-section view (bottom) are shown. (**B**) Fourier shell correlation plots for palisade maps as well as for the D13 map. Curves are given for the corrected, unmasked, masked, and masked with phase randomisations calculations.
(TIF)

**S6 Fig. Combined palisade map with C1 symmetry.** Top and side views of the combined palisade map before imposing C3 symmetry.
(TIF)

**S1 Movie. IV overview.** Tomogram of an intracellular region accumulating IV, which corresponds to the section shown in Fig 1B. IV membranes are segmented in red.
(MP4)

**S2 Movie. IMV overview.** Tomogram showing IMV in the cytoplasm, which corresponds to the section shown in Fig 1C. IMV membrane is segmented in blue, microtubules in red, and the lumen of an ER-like compartment in yellow.
(MP4)

**S3 Movie. IV assembly.** Tomogram of an IV with a pore on the viral membrane (pore labelled with magenta arrowheads). Sections of this tomograms are shown in Fig 2B.
(MOV)

**S4 Movie. Ultrastructure of IV.** Tomogram corresponding to the IV shown in Fig 3A. Scale bar = 100 nm.
(MOV)

**S5 Movie. Ultrastructure of IMV.** Tomogram of the IMV particle shown in Fig 5A. Scale bar = 100 nm.
(MOV)

**S6 Movie. Segmentation of an IMV.** The movie shows the segmentation performed on the IMV from Fig 5A. Blue is the outer layer and black labels the viral membrane, while yellow corresponds to the palisade lattice and red to the lateral bodies.
(MP4)

**S7 Movie. Ultrastructure of the naked viral core.** Tomogram of the naked core shown in Fig 6. The central magenta square marks part of the palisade, the top magenta rectangle highlights the flexible polymer observed, and the small magenta squares at the end of the movie outline the ring-like structures protruding from the palisade. Scale bar = 100 nm.
(MP4)

## Acknowledgments

We thank Alessandro Costa, Neil McDonald, Jeremy Carlton, and the Way lab (Francis Crick Institute) for insightful comments on the manuscript as well as LuYan Cao for providing the initial slide of infected cells used in our SIM analysis (Fig 2F). We also acknowledge Philip Walker and Andy Purkiss of the Structural Biology Science Technology Platform and the Scientific Computing Science Technology Platform for computational support. MHG thanks Pauline McIntosh (Francis Crick Institute) for initial guidance and training.

## Author Contributions

**Conceptualization:** Miguel Hernandez-Gonzalez, Thomas Calcraft, Andrea Nans, Peter B Rosenthal, Michael Way.

**Formal analysis:** Miguel Hernandez-Gonzalez, Thomas Calcraft, Andrea Nans.

**Funding acquisition:** Peter B Rosenthal, Michael Way.

**Investigation:** Miguel Hernandez-Gonzalez, Thomas Calcraft, Andrea Nans.

**Methodology:** Miguel Hernandez-Gonzalez.

**Supervision:** Peter B Rosenthal, Michael Way.

**Writing – original draft:** Miguel Hernandez-Gonzalez, Thomas Calcraft, Peter B Rosenthal, Michael Way.

**Writing – review & editing:** Miguel Hernandez-Gonzalez, Thomas Calcraft, Peter B Rosenthal, Michael Way.

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
