## [Editor Report · Decision Letter 0]

28 Oct 2022

Dear Michael, 

Thank you for submitting your full manuscript entitled "Cryo-ET of infected cells reveals that a succession of two lattices drives vaccinia virus assembly" after your earlier presubmission enquiry.

As you know, your manuscript has already been evaluated by the PLOS Biology editorial staff, as well as by an academic editor with relevant expertise, and we would like to send your submission out for external peer review.

Before we can send your manuscript to reviewers, we need you to complete your submission by providing the metadata that is required for full assessment. This is part of our “Easy Submission” first round process – whereby we ask that you simply upload your manuscript and cover letter with no formatting requirements. In this case, we had already evaluated your manuscript at the presubmission stage, so we have essentially skipped this step. 

To this end, please login to Editorial Manager where you will find the paper in the 'Submissions Needing Revisions' folder on your homepage. Please click 'Revise Submission' from the Action Links and complete all additional questions in the submission questionnaire.

Once your full submission is complete, your paper will undergo a series of checks in preparation for peer review. After your manuscript has passed the checks it will be sent out for review. To provide the metadata for your submission, please Login to Editorial Manager (https://www.editorialmanager.com/pbiology) within two working days, i.e. by Oct 30 2022 11:59PM.

Best wishes,

Richard

Richard Hodge, PhD

Associate Editor, PLOS Biology

rhodge@plos.org

PLOS

---

## [Decision Letter · Decision Letter 1]

20 Dec 2022

Dear Michael,

Following on from my previous e-mail, thank you for submitting your manuscript entitled "Cryo-ET of infected cells reveals that a succession of two lattices drives vaccinia virus assembly" to PLOS Biology. 

In light of the reviews, we are pleased to offer you the opportunity to address the comments from the reviewers as proposed in your rebuttal. We will then assess your revised manuscript and your response to the reviewers' comments with our Academic Editor. In addition, we would be grateful if you could address the following data and policy-related requests (A-E):

(A) We would like to suggest the following modification to the title:

“Two lattices drive vaccinia virus assembly controlling virion shape and dimensions”

(B) You may be aware of the PLOS Data Policy, which requires that all data be made available without restriction: http://journals.plos.org/plosbiology/s/data-availability. For more information, please also see this editorial: http://dx.doi.org/10.1371/journal.pbio.1001797

- Supplementary files (e.g., excel). Please ensure that all data files are uploaded as 'Supporting Information' and are invariably referred to (in the manuscript, figure legends, and the Description field when uploading your files) using the following format verbatim: S1 Data, S2 Data, etc. Multiple panels of a single or even several figures can be included as multiple sheets in one excel file that is saved using exactly the following convention: S1_Data.xlsx (using an underscore).

- Deposition in a publicly available repository. Please also provide the accession code or a reviewer link so that we may view your data before publication.

Figure 9B

(C) We ask that you please deposit the tomograms in a public data repository, such as the PDB or EMDB, and ensure that the data is publicly available. 

(D) Please also ensure that each of the relevant figure legends in your manuscript include information on *where the underlying data can be found*, and ensure your supplemental data file/s has a legend.

(E) Please ensure that your Data Statement in the submission system accurately describes where your data can be found and is in final format, as it will be published as written there. This includes provides any accession numbers for data deposited in public repositories. 

*Published Peer Review History*

*Press*

Thank you again for your submission to our journal and I am sorry again for the earlier misunderstanding. Please don't hesitate to contact us if you have any questions or comments.

Kind regards,

Richard

Richard Hodge, PhD

Associate Editor, PLOS Biology

rhodge@plos.org

PLOS

---

## [Editor Report · Decision Letter 2]

21 Dec 2022

Dear Michael,

Following on from my previous e-mail, thank you for submitting your manuscript entitled "Cryo-ET of infected cells reveals that a succession of two lattices drives vaccinia virus assembly" to PLOS Biology.

In light of the reviews, we are pleased to offer you the opportunity to address the comments from the reviewers as proposed in your rebuttal. We will then assess your revised manuscript and your response to the reviewers' comments with our Academic Editor. In addition, we would be grateful if you could address the following data and policy-related requests (A-E):

(A) We would like to suggest the following modification to the title:

“Two lattices drive vaccinia virus assembly controlling virion shape and dimensions”

(B) You may be aware of the PLOS Data Policy, which requires that all data be made available without restriction: http://journals.plos.org/plosbiology/s/data-availability. For more information, please also see this editorial: http://dx.doi.org/10.1371/journal.pbio.1001797

- Supplementary files (e.g., excel). Please ensure that all data files are uploaded as 'Supporting Information' and are invariably referred to (in the manuscript, figure legends, and the Description field when uploading your files) using the following format verbatim: S1 Data, S2 Data, etc. Multiple panels of a single or even several figures can be included as multiple sheets in one excel file that is saved using exactly the following convention: S1_Data.xlsx (using an underscore).

- Deposition in a publicly available repository. Please also provide the accession code or a reviewer link so that we may view your data before publication.

Figure 9B

(C) We ask that you please deposit the tomograms in a public data repository, such as the PDB or EMDB, and ensure that the data is publicly available.

(D) Please also ensure that each of the relevant figure legends in your manuscript include information on *where the underlying data can be found*, and ensure your supplemental data file/s has a legend.

(E) Please ensure that your Data Statement in the submission system accurately describes where your data can be found and is in final format, as it will be published as written there. This includes provides any accession numbers for data deposited in public repositories.

*Published Peer Review History*

*Press*

Thank you again for your submission to our journal and I am sorry again for the earlier misunderstanding. Please don't hesitate to contact us if you have any questions or comments.

Kind regards,

Richard

Richard Hodge, PhD

Associate Editor, PLOS Biology

rhodge@plos.org

PLOS

---

## [Editor Report · Decision Letter 3]

19 Jan 2023

Dear Michael,

Please accept my apologies for the delays you experienced at this stage of the editorial process. On behalf of my colleagues and the Academic Editor, David Bhella, I am pleased to say that we can accept your manuscript for publication, provided you address any remaining formatting and reporting issues. These will be detailed in an email you should receive within 2-3 business days from our colleagues in the journal operations team; no action is required from you until then. We will be able to formally accept your manuscript and schedule it for publication once you have completed any requested changes.

PRESS

Best wishes, 

Richard

Richard Hodge, PhD

Associate Editor, PLOS Biology

rhodge@plos.org

PLOS
